# MixSeq: Connecting Macroscopic Time Series Forecasting with Microscopic Time Series Data

**Zhibo Zhu**[*]
Ant Group
gavin.zzb@antgroup.com

**Ziqi Liu**[*]
Ant Group
ziqiliu@antgroup.com

**Ge Jin**
Ant Group
elvis.jg@antgroup.com

**Zhiqiang Zhang**
Ant Group
lingyao.zzq@antgroup.com

**Lei Chen**
Ant Group
qingli.cl@antgroup.com

**Jun Zhou**[†]
Ant Group
jun.zhoujun@antgroup.com

**Jianyong Zhou**
Ant Group
neil.zjy@antgroup.com

## Abstract

Time series forecasting is widely used in business intelligence, e.g., forecast stock market price, sales, and help the analysis of data trend. Most time series of interest are macroscopic time series that are aggregated from microscopic data. However, instead of directly modeling the macroscopic time series, rare literature studied the forecasting of macroscopic time series by leveraging data on the microscopic level. In this paper, we assume that the microscopic time series follow some unknown mixture probabilistic distributions. We theoretically show that as we identify the ground truth latent mixture components, the estimation of time series from each component could be improved because of lower variance, thus benefitting the estimation of macroscopic time series as well. Inspired by the power of Seq2seq and its variants on the modeling of time series data, we propose Mixture of Seq2seq (MixSeq), an end2end mixture model to cluster microscopic time series, where all the components come from a family of Seq2seq models parameterized by different parameters. Extensive experiments on both synthetic and real-world data show the superiority of our approach.

## 1 Introduction

Time series forecasting has proven to be important to help people manage resources and make decisions [20]. For example, probabilistic forecasting of product demand and supply in retails [9], or the forecasting of loans [1] in a financial institution can help people do inventory or financing planning to maximize the profit. Most time series of interest are macroscopic time series, e.g., the sales of an online retail platform, the loans of a financial institution, or the number of infections caused by some pandemic diseases in a state, that are comprised of microscopic time series, e.g., the sales of a merchant in the online retail, the loans from a customer given the financial institution, or the number of infections in a certain region. That is, the observed macroscopic time series are just the aggregation or sum of microscopic time series.

---

[*]Equal contribution.
[†]Corresponding author.

35th Conference on Neural Information Processing Systems (NeurIPS 2021).

Although various time series forecasting models, e.g., State Space Models (SSMs) [13], Autoregressive (AR) models [2], or deep neural networks [5], have been widely studied for decades, all of them study the modeling of time series without considering the connections between macroscopic time series of interest and the underlying time series on the microscopic level.

In this paper, we study the question whether the forecasting of macroscopic time series can be improved by leveraging the underlying microscopic time series, and the answer is yes. Basically, though accurately modeling each microscopic time series could be challenging due to large variations, we show that by carefully clustering microscopic time series into clusters, i.e., clustered time series, and using canonical approaches to model each of clusters, finally we can achieve promising results by simply summing over the forecasting results of each cluster.

To be more specific, **first**, we assume that the microscopic time series are generated from a probabilistic mixture model [24] where there exist $K$ components. The generation of each microscopic time series is by first selecting a component $z$ from $\{1, ..., K\}$ with a prior $p(z)$ (a Discrete distribution), then generating the microscopic observation from a probabilistic distribution $p(x; \Phi_z, z)$ parameterized by the corresponding component $\Phi_z$. We show that as we can identify the ground truth components of the mixture, and the ground truth assignment of each microscopic observation, independent modeling of time series data from each component could be improved due to lower variance, and further benefitting the estimation of macroscopic time series that are of interest. **Second**, inspired by recent successes of Seq2seq models [36, 10, 12] based on deep neural networks, e.g., variants of recurrent neural networks (RNNs) [16, 41, 22], convolutional neural networks (CNNs) [4, 15], and Transformers [19, 38], we propose Mixture of Seq2seq (MixSeq), a mixture model for time series, where the components come from a family of Seq2seq models parameterized by different parameters. **Third**, we conduct synthetic experiments to demonstrate the superiority of our approach, and extensive experiments on real-world data to show the power of our approach compared with canonical approaches.

**Our contributions**. We summarize our contributions in two-fold. **(1)** We show that by transforming the original macroscopic time series via clustering, the expected variance of each clustered time series could be optimized, thus improving the accuracy and robustness for the estimation of macroscopic time series. **(2)** We propose MixSeq which is an end2end mixture model with each component coming from a family of Seq2seq models. Our empirical results based on MixSeq show the superiority compared with canonical approaches.

## 2  Background

In this section, we first give a formal problem definition. We then review the bases related to this work, and have a discussion of related works.

**Problem definition**. Let us assume a macroscopic time series $x_{1:t_0} = [x_1, ..., x_{t_0}]$, and $x_t \in \mathbb{R}$ denotes the value of time series at time $t$. We aim to predict the next $\tau$ time steps, i.e., $x_{t_0+1:t_0+\tau}$. We are interested in the following conditional distribution

$$p(x_{t_0+1:t_0+\tau}|x_{1:t_0}) = \prod_{t=t_0+1}^{t_0+\tau} p(x_t|x_{<t}; \Theta), \tag{1}$$

where $x_{<t}$ represents $x_{1:t-1}$ in interval $[1, t)$. To study the above problem, we assume that the macroscopic time series is comprised of $m$ microscopic time series, i.e., $x_t = \sum_{i=1}^{m} x_{i,t}$ where $x_{i,t} \in \mathbb{R}$ denotes the value of the $i$-th microscopic time series at time $t$. We aim to cluster the $m$ microscopic time series into $K$ clustered time series $\left\{x_{1:t_0}^{(z)}\right\}_{z=1}^{K}$, where $x_t^{(z)} = \sum_{\{i|z_i=z, \forall i\}} x_{i,t}$ given the label assignment of the $i$-th microscopic time series $z_i \in \{1, ..., K\}$. This is based on our results in Section 3 that the macroscopic time series forecasting can be improved with optimal clustering. Hence, instead of directly modeling $p(x_{t_0+1:t_0+\tau})$, we study the clustering of $m$ microscopic time series in Section 4, and model the conditional distribution of clustered time series $\left\{p(x_{t_0+1:t_0+\tau}^{(z)})\right\}_{z=1}^{K}$ with canonical approaches.

## 2.1 Seq2seq: encoder-decoder architectures for time series

An encoder-decoder based neural network models the conditional distribution Eq. (1) as a distribution from the exponential families, e.g., Gaussian, Gamma or Binomial distributions, with sufficient statistics generated from a neural network. The encoder feeds $x_{<t}$ into a neural architecture, e.g., RNNs, CNNs or self-attentions, to generate the representation of historical time series, denoted as $h_t$, then we use a decoder to yield the result $x_t$. After $\tau$ iterations in an autoregressive style, it finally generates the whole time series to be predicted.

To instantiate above Seq2seq architecture, we denote $o_{1:t}$, where $o_t \in \mathbb{R}^d$, as covariates that are known a priori, e.g., dates. We denote $Y_t = [x_{1:t-1} \| o_{2:t}] \in \mathbb{R}^{(t-1)\times(d+1)}$ where we use $\|$ for concatenation. The encoder generates the representation $h_t$ of $x_{<t}$ via Transformer [36, 19] as follows. We first transform $Y_t$ by some functions $\rho(\cdot)$, e.g., causal convolution [19] to $H^{(0)} = \rho(Y_t) \in \mathbb{R}^{(t-1)\times d_k}$. Transformer then iterates the following self-attention layer $L$ times:

$$H^{(l)} = \text{MLP}^{(l)}(H^{(\text{tmp})}), \ \ H^{(\text{tmp})} = \text{SOFTMAX}\left(\frac{Q^{(l)}K^{(l)\top}}{\sqrt{d_q}}M\right)V^{(l)},$$

$$Q^{(l)} = H^{(l-1)}W_q^{(l)}, K^{(l)} = H^{(l-1)}W_k^{(l)}, V^{(l)} = H^{(l-1)}W_v^{(l)}. \tag{2}$$

That is, we first transform $Y^3$ into query, key, and value matrices, i.e., $Q = YW_q$, $K = YW_k$, and $V = YW_v$ respectively, where $W_q \in \mathbb{R}^{d_k \times d_q}, W_k \in \mathbb{R}^{d_k \times d_q}, W_v \in \mathbb{R}^{d_k \times d_v}$ in each layer are learnable parameters. Then we do scaled inner product attention to yield $H^{(l)} \in \mathbb{R}^{(t-1)\times d_k}$ where $M$ is a mask matrix to filter out rightward attention by setting all upper triangular elements to $-\infty$. We denote $\text{MLP}(\cdot)$ as a multi-layer perceptron function. Afterwards, we can generate the representation $h_t \in \mathbb{R}^{d_p}$ for $x_{<t}$ via $h_t = \nu(H^{(L)})$ where we denote $\nu(\cdot)$ as a deep set function [42] that operates on rows of $H^{(L)}$, i.e., $\nu(\{H_1^{(L)}, ..., H_{t-1}^{(L)}\})$. We denote the feedforward function to generate $H^{(L)}$ as $H^{(L)} \sim g(H^{(0)})$, i.e., Eq (2).

Given $h_t$, the decoder generates the sufficient statistics and finally yields $x_t \sim p(x; \text{MLP}(h_t))$ from a distribution in the exponential family.

## 2.2 Related works

**Time series forecasting** has been studied for decades. We summarize works related to time series forecasting into two categories. **First**, many models come from the family of autoregressive integrated moving average (ARIMA) [7, 2], where AR indicates that the evolving variable of interest is regressed on its own lagged values, the MA indicates that the regression error is actually a linear combination of error terms, and the "I" indicates that the data values have been replaced with the difference between their values and the previous values to handle non-stationary [27]. The State Space Models (SSM) [13] aim to use state transition function to model the transfer of states and generate observations via a observation function. These statistical approaches typically model time series independently, and most of them only utilize values from history but ignore covariates that are important signals for forecasting. **Second**, as rapid development of deep neural networks, people started studying many neural networks for the modeling of time series [5, 20]. Most successful neural networks are based on the encoder-decoder architectures [36, 10, 12, 33, 3, 11, 20, 21], namely Seq2seq. Basically, various Seq2seq models based on RNNs [16, 32, 37, 18, 41, 22], CNNs [4, 15], and Transformers (self-attentions) [19, 38] are proposed to model the non-linearity for time series.

No matter models studied in statistics or deep neural networks, these works mainly focus on the forecasting of single or multivariate time series, but ignore the auxiliary information that the time series could be made up of microscopic data.

**Time series clustering** is another topic for exploratory analysis of time series. We summarize the literature into three categories, i.e., study of distance functions, generative models, and feature extraction for time series. **First**, Dynamic time wrapping [28], similarity metric that measures temporal dynamics [40], and specific measures for the shape [26] of time series are proposed to adapt to various time series characteristics, e.g., scaling and distortion. Typically these distance functions are mostly manually defined and cannot generalize to more general settings. **Second**, generative

---

[3] We ignore the subscript for simplicity in condition that the context is of clarity.

model based approaches assume that the observed time series is generated by an underlying model, such as hidden markov model [25] or mixture of ARMA [39]. **Third**, early studies on feature extraction of time series are based on component analysis [14], and kernels, e.g., u-shapelet [43]. As the development of deep neural networks, several encoder-decoder architectures [23, 21] are proposed to learn better representations of time series for clustering.

However, the main purpose of works in this line is to conduct exploratory analysis of time series, while their usage for time series forecasting has never been studied. That is, these works define various metrics to evaluate the goodness of the clustering results, but how to learn the optimal clustering for time series forecasting remains an open question.

## 3 Microscopic time series under mixture model

We analyze the variance of mixture models, and further verify our results with simple toy examples.

### 3.1 Analyses on the variance of mixture model

In this part, we analyze the variance of probabilistic mixture models. A mixture model [24] is a probabilistic model for representing the presence of subpopulations within an overall population. Mixture model typically consists of a prior that represents the probability over subpopulations, and components, each of which defines the probability distribution of the corresponding subpopulation. Formally, we can write

$$f(x) = \sum_i p_i \cdot f_i(x), \tag{3}$$

where $f(\cdot)$ denotes the mixture distribution, $p_i$ denotes the prior over subpopulations, and $f_i(\cdot)$ represents the distribution corresponding to the $i$-th component.

**Proposition 1.** *Assuming the mixture model with probability density function $f(x)$, and corrrresponding components $\{f_i(x)\}_{i=1}^{K}$ with constants $\{p_i\}_{i=1}^{K}$ ($\{p_i\}_{i=1}^{K}$ lie in a simplex), we have $f(x) = \sum_i p_i f_i(x)$. In condition that $f(\cdot)$ and $\{f_i(\cdot)\}_{i=1}^{K}$ have first and second moments, i.e., $\mu^{(1)}$ and $\mu^{(2)}$ for $f(x)$, and $\left\{\mu_i^{(1)}\right\}_{i=1}^{K}$ and $\left\{\mu_i^{(2)}\right\}_{i=1}^{K}$ for components $\{f_i(x)\}_{i=1}^{K}$, we have:*

$$\sum_i p_i \cdot \mathrm{Var}(f_i) \leq \mathrm{Var}(f). \tag{4}$$

We use the fact that $\mu^{(k)} = \sum_i p_i \mu_i^{(k)}$. By using Jensen's Inequality on $\sum_i p_i \left(\mu_i^{(1)}\right)^2 \geq \left(\sum_i p_i \mu_i^{(1)}\right)^2$, we immediately yield the result. See detailed proofs in supplementary.

This proposition states that, if we have limited data samples (always the truth in reality) and in case we know the ground truth data generative process a priori, i.e., the exact generative process of each sample from its corresponding component, the variance on expectation conditioned on the ground truth data assignment should be no larger than the variance of the mixture. Based on the assumption that microscopic data are independent, the variance of the aggregation of clustered data should be at least no larger than the aggregation of all microscopic data, i.e., the macroscopic data. So the modeling of clustered data from separate components could possibly be more accurate and robust compared with the modeling of macroscopic data. This result motivates us to forecast macroscopic time series by clustering the underlying microscopic time series. Essentially, we transform the original macroscopic time series data to clusters with lower variances using a clustering approach, then followed by any time series models to forecast each clustered time series. After that, we sum over all the results from those clusters so as to yield the forecasting of macroscopic time series. We demonstrate this result with toy examples next.

### 3.2 Demonstration with toy examples

We demonstrate the effectiveness of forecasting macroscopic time series by aggregating the forecasting results from clustered time series.

Table 1: We run the experiments 5 times, and show the average results (SMAPE) of macro results and clustered results with ground truth clusters. Lower is better.

| | GP time series data | | ARMA time series data | |
|---|---|---|---|---|
| | 3 clusters | 5 clusters | 3 clusters | 5 clusters |
| macro results | 0.0263 | 0.0242 | 0.5870 | 0.5940 |
| clustered results | **0.0210** | **0.0198** | **0.3590** | **0.3840** |

**Simulation setting.** We generate microscopic time series from a mixture model, such as Gaussian process (GP) [31] or ARMA [8] with 3 or 5 components. We generate 5 time series for each component, and yield 15 or 25 microscopic time series in total. We sum all the time series as the macroscopic time series. We get clustered time series by simply summing microscopic time series from the same component. Our purpose is to compare the performance between forecasting results directly on macroscopic time series (macro results) and sum of forecasting results of clustered time series (clustered results). We set the length of time series as 360, and use rolling window approach for training and validating our results in the last 120 time steps (i.e., at each time step, we train the model using the time series before current time point, and validate using the following 30 values). We fit the data with either GP or ARMA depending on the generative model. We describe the detailed simulation parameters of mixture models in supplementary.

**Simulation results.** Table 1 shows the results measured by symmetric mean absolute percentage error (SMAPE)[4]. It is obvious that no matter time series generated by mixture of GP or mixture of ARMA, the clustered results are superior to macro results. In other words, if we knew the ground truth component of each microscopic time series, modeling on clustered data aggregated by time series from the same component would have better results compared with directly modeling the macroscopic time series.

## 4 MixSeq: a mixture model for time series

Based on the analysis in Section 3, we assume microscopic time series follow a mixture distribution, and propose a mixture model, MixSeq, to cluster microscopic time series.

### 4.1 Our model

Our model assumes that each of $m$ microscopic time series follows a mixture probabilistic distribution. To forecast $x_{t_0+1:t_0+\tau}$ given $x_{1:t_0}$, our approach is to first partition $\{x_{i,1:t_0}\}_{i=1}^m$ into $K$ clusters via MixSeq. Since the distribution is applicable to all microscopic time series, we ignore the subscript $i$ and time interval $1:t_0$ for simplicity. We study the following generative probability of $x \in \mathbb{R}^{t_0}$:

$$p(x) = \sum_z p(x, z) = \sum_z p(z)p(x|z) = \sum_z p(z) \prod_{t=1}^{t_0} p(x_t|x_{<t}, z; \Phi_z), \qquad (5)$$

where $z \in \{1, 2, ..., K\}$ is the discrete latent variable, $K$ is the number of components in the mixture model, $p(z)$ is the prior of cluster indexed by $z$, and $p(x|z)$ is the probability of time series $x$ generated by the corresponding component governed by parameter $\Phi_z$. Note that we have $K$ parameters $\Theta = \{\Phi_1, \Phi_2, \ldots, \Phi_K\}$ in the mixture model.

We use ConvTrans introduced in [19] as our backbone to model the conditional $p(x_t|x_{<t}, z; \Phi_z)$. To instantiate, we model the conditional $p(x_t|x_{<t}, z; \Phi_z)$ by first generating $H^{(0)} \sim \rho(Y_t)$ via causal convolution [19]. We then generate $H^{(L)} \sim g(H^{(0)}) \in \mathbb{R}^{(t-1) \times d_k}$ given $x_{<t}$. Finally we generate the representation for $x_{<t}$ as $h_t = \nu(H^{(L)}) = \sigma(W_s \sum_{j=1}^{t-1} H_j^{(L)}) \in \mathbb{R}^{d_p}$, where $W_s \in \mathbb{R}^{d_p \times d_k}$ and $\sigma$ as ReLU activation function. Afterwards, we decode $h_t$ to form the specific distribution from an exponential family. In particular, we use Gaussian distribution $p(x_t|x_{<t}, z; \Phi_z) = \mathcal{N}(x_t; \mu_t, \sigma_t^2)$, where the mean and variance can be generated by following transformations,

$$\mu_t = w_\mu^T h_t + b_\mu, \quad \sigma_t^2 = \log(1 + \exp(w_\sigma^T h_t + b_\sigma)), \qquad (6)$$

---

[4]Details are in supplementary

where $w_\mu, w_\sigma \in \mathbb{R}^{d_p}$ are parameters, and $b_\mu, b_\sigma \in \mathbb{R}$ are biases.

## 4.2 Posterior inference and learning algorithms

We aim to learn the parameter $\Theta$ and efficiently infer the posterior distribution of $p(z|x)$ in Eq. (5). However, it is intractable to maximize the marginal likelihood $p(x)$ after taking logarithm, i.e., $\log p(x)$, since of the involvement of logarithm of sum. To tackle this non-convex problem, we resort to stochastic auto-encoding variational Bayesian algorithm (AEVB) [17]. Regarding single microscopic time series, the variational lower bound (LB) [17] on the marginal likelihood is as below.

$$
\begin{aligned}
\log p(x) = \log \sum_z p(x, z) &\geq \sum_z q(z|x) \log \frac{p(x, z)}{q(z|x)} \\
&= \mathbb{E}_{q(z|x)} \log p(x|z) - \mathrm{KL}\left(q(z|x)\|p(z)\right) = \mathrm{LB},
\end{aligned}
\tag{7}
$$

where $q(z|x)$ is the approximated posterior of the latent variable $z$ given time series $x$. The benefit of using AEVB is that we can treat $q(z|x)$ as an encoder modeled by a neural network. Hence, we reuse the ConvTrans [19] as our backbone, and model $q(z|x)$ as:

$$
q(z|x) = \mathrm{SOFTMAX}(W_a \cdot \nu(H^{(L)})), \quad H^{(L)} = g(\rho(Y_{t_0})),
\tag{8}
$$

where we denote $Y_{t_0} = [x_{1:t_0} \| o_{1:t_0}] \in \mathbb{R}^{t_0 \times (d+1)}$, $\nu(H^{(L)}) = \sigma(W_s \cdot \sum_{j=1}^{t_0} H_j^{(L)})$ with parameter $W_s \in \mathbb{R}^{d_p \times d_k}$ is the deep set function, and $W_a \in \mathbb{R}^{K \times d_p}$ as parameters to project the encoding to $K$ dimension. After the softmax operator, we derive the posterior distribution that lies in a simplex of $K$ dimension. Note that we use distinct $\rho(\cdot)$'s, $g(\cdot)$'s and $\nu(\cdot)$'s with different parameters to model $q(z|x)$ and $\{p(x_t|x_{<t}, z)\}_{z=1}^K$ respectively. We assign each microscopic $x_i$ to cluster $z_i = \arg\max_z q(z|x_i)$ in our experiments.

**Mode collapsing.** We find that directly optimizing the lower bound in Eq. (7) suffers from the mode collapsing problem. That is, the encoder $q(z|x)$ tends to assign all microscopic time series to one cluster, and does not effectively distinguish the data as expected, thus implying $I(x; z) = 0$ ($I(\cdot)$ for mutual information). In order to address the above mode collapsing problem, we add $I(x; z)$ to the lower bound in Eq. (7) which expects that the latent variable $z$ can extract discriminative information from different time series [44]. Then, we have

$$
\begin{aligned}
&\mathbb{E}_x(\mathbb{E}_{q(z|x)} \log p(x|z)) - \mathbb{E}_x(\mathrm{KL}(q(z|x)\|p(z))) + I(x; z) \\
&= \mathbb{E}_x(\mathbb{E}_{q(z|x)} \log p(x|z)) - \mathrm{KL}(q(z)\|p(z)),
\end{aligned}
\tag{9}
$$

where $q(z) = \frac{1}{m} \sum_{i=1}^m q(z|x_i)$ is an average of approximated posteriors over all microscopic data. We approximate this term by using a mini-batch of $m'$ samples, i.e., $q(z) = \frac{1}{m'} \sum_{i' \in \mathcal{B}} q(z|x_{i'})$.

**Annealing tricks.** Regarding long-length time series, the reconstruction loss and KL divergence in Eq. (9) are out of proportion. In this situation, the KL divergence has few effects on the optimization objective. So we finally derive the following objective to maximize:

$$
\mathbb{E}_x(\mathbb{E}_{q(z|x)} \log p(x|z)) - \alpha \cdot \mathrm{KL}(q(z)\|p(z)) - \lambda \cdot \|\Theta\|
\tag{10}
$$

where $\alpha$ is the trade-off hyperparameter. We use the following annealing strategy $\alpha = \max(a, b \times e^{(-\beta n)})$ to dynamically adjust $\alpha$ in the training process, where $\beta$ is the parameter controlling the rate of descent. Meanwhile, we also involve the $\ell_2$-norm regularizers on Seq2seq's parameters $\Theta$ with hyperparameter $\lambda \geq 0$.

# 5 Experimental results

We conduct extensive experiments to show the advantage of MixSeq. We evaluate the clustering performance of MixSeq on synthetic data, present the results of macroscopic time series forecasting on real-world data, and analyze the sensitivity of the cluster number of MixSeq.

## 5.1 Synthetic datasets

To demonstrate MixSeq's capability of clustering microscopic time series that follow various probabilistic mixture distributions, we conduct clustering experiments on synthetic data with ground truth.

Table 2: Mean and standard deviation (SD, in bracket) of Rand Index (RI, the higher the better) by clustering on synthetic data generated by ARMA and DeepAR. MixSeq-infer represents that we infer the cluster of new data generated by different models after training MixSeq. On ARMA data, MixSeq and MixARMA have comparable performance; on DeepAR data, MixARMA degrades significantly which shows the effectiveness of MixSeq.

|  | ARMA synthetic data | | DeepAR synthetic data | |
|  | 2 clusters | 3 clusters | 2 clusters | 3 clusters |
| --- | --- | --- | --- | --- |
| MixARMA | **0.9982**(0.0001) | 0.9509(0.1080) | 0.7995(0.2734) | 0.7687(0.0226) |
| MixSeq | 0.9915(0.0024) | **0.9540**(0.0974) | **0.9986**(0.0003) | **0.8460**(0.0774) |
| MixSeq-infer | 0.9929(0.0027) | 0.9544(0.0975) | 0.9982(0.0006) | 0.8460(0.0775) |

We generate two kinds of synthetic time series by ARMA [8] and DeepAR [32] respectively. For each model, we experiment with different number of clusters (2 and 3) generated with components governed by different parameters.

**Experiment setting.** To generate data from ARMA, we use ARMA(2, 0) and $x_t = \phi_1 x_{t-1} + \phi_2 x_{t-2} + \epsilon_t$ with $\epsilon_t \sim N(0, 0.27)$. We set parameters $[\phi_1, \phi_2]$ for three components as $[-0.25, 0.52]$, $[0.34, 0.27]$, and $[1.5, -0.75]$ respectively. The synthetic time series from a mixture of 2 components are generated using the first two components. The synthetic time series from a mixture of 3 components are generated using all 3 components. To generate data from DeepAR, we use the DeepAR model with one LSTM layer, and the hidden number of units is 16. Since it is difficult to randomly initialize the parameters of DeepAR, we train a base model on the real-world Wiki dataset [35] (discussed in section 5.2). To build the other two DeepAR components, we respectively add random disturbance $\mathcal{N}(0, 0.01)$ to the parameters of the base model. For each cluster, we generate $10,000$ time series with random initialized sequences, and set the length of time series as $100$.

We use 1-layer causal convolution Transformer (ConvTrans [19]) as our backbone model in MixSeq. We use the following parameters unless otherwise stated. We set the number of multi-heads as 2, kernel size as 3, the number of kernel for causal convolution $d_k = 16$, dropout rate as 0.1, the penalty weight on the $\ell_2$-norm regularizer as 1e-5, and $d_p = d_v = 16$. Meanwhile, we set the prior $p(z)$ as $1/K$, $\forall z$. For the training parameters, we set the learning rate as 1e-4, batch size as 256 and epochs as 100. Furthermore, the $\alpha$ in MixSeq is annealed using the schedule $\alpha = \max(5, 20e^{(-0.03n)})$, where $n$ denotes the current epoch, and $\alpha$ is updated in the $[10, 30, 50]$-th epochs. For comparison, we employ MixARMA [39], a mixture of ARMA(2, 0) model optimized by EM algorithm [6], as our baseline. Both methods are evaluated using Rand Index (RI) [29] (more details in supplementary).

**Experiment results.** We show the clustering performance of MixSeq and MixARMA on the synthetic data in Table 2. The results are given by the average of 5 trials. Regarding the synthetic data from ARMA, both MixSeq and MixARMA perform very well. However, for the synthetic data from DeepAR, MixARMA degrades significantly while MixSeq achieves much better performance. This suggests that MixSeq can capture the complex nonlinear characteristics of time series generated by DeepAR when MixARMA fails to do so. Furthermore, we also generate new time series by the corresponding ARMA and DeepAR models, and infer their clusters with the trained MixSeq model. The performance is comparable with the training performance, which demonstrates that MixSeq actually captures the generation mode of time series.

## 5.2 Real-world datasets

We further evaluate the effectiveness of our model on the macroscopic time series forecasting task. We compare MixSeq with existing clustering methods and state-of-the-art time series forecasting approaches on several real-world datasets. Specifically, for each dataset, the goal is to forecast the macroscopic time series aggregated by all microscopic data. We cluster microscopic time series into groups, and aggregate the time series in each group to form the clustered time series. Then, we train the forecasting models on the clustered time series separately, and give predictions of each clustered time series. Finally, the estimation of macroscopic time series is obtained by aggregating all the predictions of clustered time series.

Table 3: Real-world dataset summary.

| dataset | # microscopic time series | length of time series | train interval | test internal |
|---|---|---|---|---|
| Rossmann | 1115 | 942 | 20130101-20141231 | 20150101-20150731 |
| M5 | 30490 | 1941 | 20110129-20160101 | 20160101-20160619 |
| Wiki | 309765 | 1827 | 20150701-20191231 | 20200101-20200630 |

Table 4: Comparisons on the microscopic time series clustering methods for macroscopic time series forecasting combined with three network-based forecasting methods: testing $R_{0.5}/R_{0.9}$-loss on three real-world datasets. Lower is better.

| | | Macro | DTCR | MixARMA | MixSeq |
|---|---|---|---|---|---|
| Rossmann | DeepAR | 0.1904/**0.0869** | 0.2292/0.1432 | 0.1981/0.1300 | **0.1857**/0.0987 |
| | TCN | 0.1866/0.1005 | 0.2023/0.1633 | 0.1861/0.1160 | **0.1728/0.0997** |
| | ConvTrans | 0.1861/0.0822 | 0.2077/0.0930 | 0.1866/0.0854 | **0.1847/0.0813** |
| M5 | DeepAR | **0.0548/0.0289** | 0.0787/0.0627 | 0.0624/0.0582 | 0.0582/0.0445 |
| | TCN | 0.0790/0.0635 | 0.0847/0.0805 | 0.0762/0.0789 | **0.0694/0.0508** |
| | ConvTrans | 0.0553/0.0260 | 0.0514/0.0260 | 0.0497/0.0257 | **0.0460/0.0238** |
| Wiki | DeepAR | 0.0958/0.0962 | 0.1073/0.1336 | 0.0974/0.1070 | **0.0939/0.0901** |
| | TCN | 0.0966/0.1064 | 0.1237/0.1480 | 0.0963/0.1218 | **0.0886/0.0980** |
| | ConvTrans | 0.0968/0.0589 | 0.1029/0.0531 | 0.0961/0.0594 | **0.0901/0.0516** |

We report results on three real-world datasets, including Rossmann[5], M5[6] and Wiki [35]. The Rossmann dataset consists of historical sales data of $1,115$ Rossmann stores recorded every day. Similarly, the M5 dataset consists of $30,490$ microscopic time series as the daily sales of different products in ten Walmart stores in USA. The Wiki dataset contains $309,765$ microscopic time series representing the number of daily views of different Wikipedia articles. The dataset summary is shown in Table 3, together with the setting of data splits.

**Experiment setting.** We summarize the clustering strategies for macroscopic time series forecasting as follows. **(1)** "DTCR" [21] is the deep temporal clustering representation method which integrates the temporal reconstruction, K-means objective and auxiliary classification task into a single Seq2seq model. **(2)** "MixARMA" [39] is the mixture of ARMA model that uses ARMA to capture the characteristics of microscopic time series. **(3)** "MixSeq" is our model with 1-layer causal convolution Transformer [19]. **(4)** We also report the results that we directly build forecasting model on the macroscopic time series without leveraging the microscopic data, named as "Macro".

For time series forecasting, we implement five methods combined with each clustering strategy, including ARMA [8], Prophet [34], DeepAR [32], TCN [4], and ConvTrans [19]. ARMA and Prophet give the prediction of point-wise value for time series, while DeepAR, TCN and ConvTrans are methods based on neural network for probabilistic forecasting with Gaussian distribution. We use the rolling window strategy on the test interval, and compare different methods in terms of the long-term forecasting performance for 30 days. The data of last two months in train interval are used as validation data to find the optimal model.

We do grid search for the following hyperparameters in clustering and forecasting algorithms, i.e., the number of clusters $\{3, 5, 7\}$, the learning rate $\{0.001, 0.0001\}$, the penalty weight on the $\ell_2$-norm regularizers $\{1e-5, 5e-5\}$, and the dropout rate $\{0, 0.1\}$. The model with best validation performance is applied for obtaining the results on test interval. Meanwhile, we set batch size as 128, and the number of training epochs as 300 for Rossmann, 50 for M5 and 20 for Wiki. For DTCR, we use the same setting as [21].

Regarding time series forecasting models, we apply the default setting to ARMA and Prophet provided by the Python packages. The architectures of DeepAR, TCN and ConvTrans are as follows.

---

[5]https://www.kaggle.com/c/rossmann-store-sales
[6]https://www.kaggle.com/c/m5-forecasting-accuracy

Table 5: Comparisons on the microscopic time series clustering methods for macroscopic time series forecasting combined with five forecasting methods: testing SMAPE on three real-world datasets.

|  |  | Macro | DTCR | MixARMA | MixSeq |
|---|---|---|---|---|---|
| Rossmann | ARMA | 0.2739(0.0002) | 0.2735(0.0106) | 0.2736(0.0013) | **0.2733**(0.0012) |
|  | Prophet | 0.1904(0.0007) | **0.1738**(0.0137) | 0.1743(0.0037) | 0.1743(0.0026) |
|  | DeepAR | 0.1026(0.0081) | 0.1626(0.0117) | 0.1143(0.0088) | **0.0975**(0.0013) |
|  | TCN | 0.1085(0.0155) | 0.1353(0.0254) | 0.1427(0.0180) | **0.1027**(0.0075) |
|  | ConvTrans | 0.1028(0.0091) | 0.1731(0.0225) | 0.1022(0.0041) | **0.0961**(0.0019) |
| M5 | ARMA | **0.0540**(0.0001) | 0.0544(0.0018) | 0.0541(0.0003) | 0.0543(0.0001) |
|  | Prophet | 0.0271(0.0003) | 0.0271(0.0003) | 0.0269(0.0002) | **0.0267**(0.0002) |
|  | DeepAR | **0.0278**(0.0034) | 0.0410(0.0046) | 0.0319(0.0063) | 0.0298(0.0029) |
|  | TCN | 0.0412(0.0075) | 0.0447(0.0044) | 0.0395(0.0094) | **0.0358**(0.0014) |
|  | ConvTrans | 0.0274(0.0048) | 0.0253(0.0020) | 0.0245(0.0024) | **0.0227**(0.0006) |
| Wiki | ARMA | **0.0362**(0.0001) | 0.0363(0.0006) | 0.0364(0.0005) | **0.0362**(0.0002) |
|  | Prophet | **0.0413**(0.0001) | 0.0423(0.0008) | 0.0434(0.0003) | 0.0420(0.0005) |
|  | DeepAR | 0.0481(0.0008) | 0.0552(0.0015) | 0.0489(0.0006) | **0.0470**(0.0002) |
|  | TCN | 0.0494(0.0076) | 0.0654(0.0022) | 0.0491(0.0015) | **0.0446**(0.0023) |
|  | ConvTrans | 0.0471(0.0029) | 0.0497(0.0012) | 0.0466(0.0001) | **0.0440**(0.0010) |

The number of layers and hidden units are 1 and 16 for DeepAR. The number of multi-heads and kernel size are 2 and 3 for ConvTrans. The kernel size is 3 for TCN with dilations in $[1, 2, 4, 8]$. We also set batch size as 128 and the number of epochs as 500 for all forecasting methods.

**Experiment results.** Following [19, 30, 35], we evaluate the experimental methods using SMAPE and $\rho$-quantile loss $R_\rho$[7] with $\rho \in (0, 1)$. The SMAPE results of all combination of clustering and forecasting methods are given in Table 5. Table 4 shows the $R_{0.5}/R_{0.9}$-loss for DeepAR, TCN and ConvTrans which give probabilistic forecasts. All results are run in 5 trials. The best performance is highlighted by bold character. We observe that MixSeq is superior to other three methods, suggesting that clustering microscopic time series by our model is able to improve the estimation of macroscopic time series. Meanwhile, Macro and MixARMA have comparable performance and are better than DTCR, which further demonstrates the effectiveness of our method, i.e., only proper clustering methods are conductive to macroscopic time series forecasting.

## 5.3 Sensitivity analysis of cluster number

The cluster number $K$ is a critical hyperparameter of MixSeq. To analyze its effect on the forecasting of macroscopic time series, we conduct experiments on both synthetic data and real-world data. The results state the importance of setting a proper number of clusters. We suggest do binary search on this critical hyperparameter. Details are as follows.

**Synthetic data.** Following the experimental setting in section 5.1, we generate $10,000$ microscopic time series from 3 different parameterized ARMA respectively. That is, the ground truth number of clusters is 3 and there are $30,000$ microscopic samples in total. The aggregation of all samples is the macroscopic time series which is the forecasting target of interest. Then, we compare the forecasting performance between the method that directly forecasts macroscopic time series (denoted as Macro) and our method with different cluster numbers (including 2, 3, 5, denoted as MixSeq_2, MixSeq_3 and MixSeq_5 respectively). We fix the forecasting method as ARMA, and apply the rolling window approach for T+10 forecasting in the last 40 time steps. The average SMAPE of 5 trials are $0.807$, $0.774$, $0.731$ and $0.756$ for Macro, MixSeq_2, MixSeq_3 and MixSeq_5 respectively. MixSeq_3 being set with ground truth cluster number shows the best performance, while MixSeq_2 and MixSeq_5 would degenerate though still better than Macro. This result shows the importance of setting a proper number of clusters.

**Real-world data.** We do the evaluation on three real-world datasets by varying the cluster number $K$ of MixSeq while maintaining the other parameters fixed. For Rossmann and M5 datasets, we set the

---

[7]Detailed definition is in supplementary.

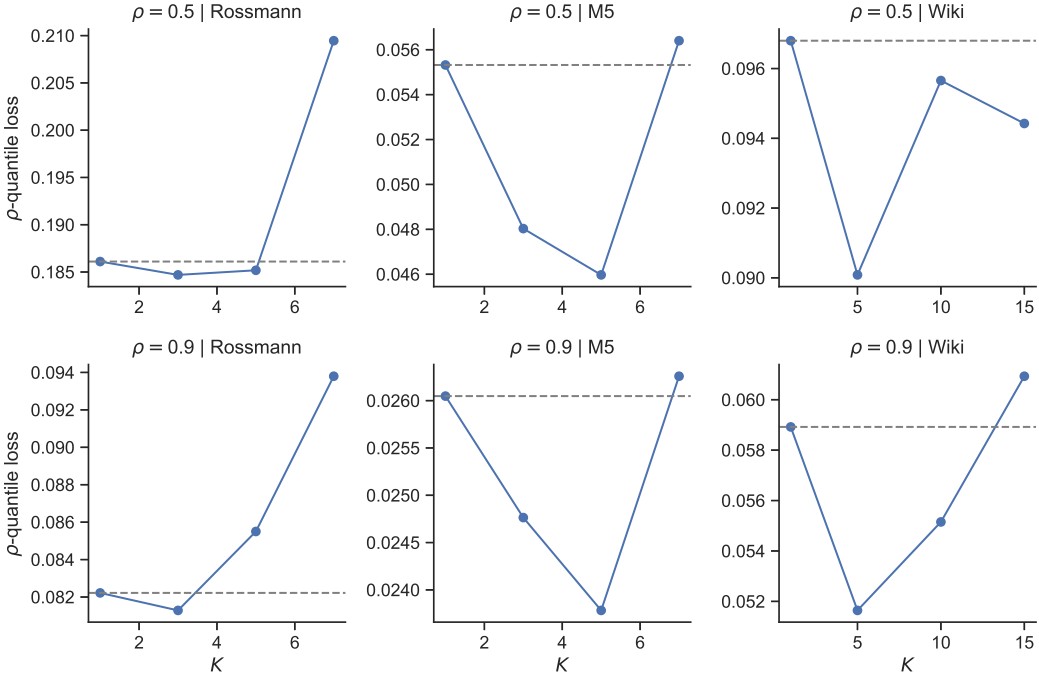

Figure 1: The macroscopic time series forecasting performance based on MixSeq with different cluster number $K$ on three real-world datasets. The time series forecasting method is fixed as causal convolution Transformer. Top three figures show the $R_{0.5}$-loss and bottom three figures show the $R_{0.9}$-loss.

cluster number $K \in \{3, 5, 7\}$, while we explore the cluster number $K \in \{5, 10, 15\}$ on Wiki dataset. The architecture and training parameters of MixSeq are same to section 5.2, except that we set the dropout rate as $0.1$, the penalty weight on the $\ell_2$-norm regularizer as 5e-5, and the learning rate as 1e-4. Meanwhile, we also fix the time series forecasting method as causal convolution Transformer (ConvTrans).

Figure 1 reports the macroscopic time series forecasting performance (testing on $R_{0.5}$ and $R_{0.9}$ loss) based on MixSeq with different cluster number $K$ on three real-world datasets. The horizontal dashed lines are the results with $K = 1$ that directly building ConvTrans model on the macroscopic time series without leveraging the microscopic data (named as "Macro" in section 5.2). It is obvious that each dataset has its own suitable number of clusters, and our method is relatively sensitive to $K$, especially on the dataset with less microscopic time series, such as Rossmann. Similar to the accurately modeling of each microscopic time series, the larger cluster number $K$ of MixSeq also brings large variance to macroscopic time series forecasting, which degrades the performance of our method.

## 6   Conclusion

In this paper, we study the problem that whether macroscopic time series forecasting can be improved by leveraging microscopic time series. Under mild assumption of mixture models, we show that appropriately clustering microscopic time series into groups is conductive to the forecasting of macroscopic time series. We propose MixSeq to cluster microscopic time series, where all the components come from a family of Seq2seq models parameterized with different parameters. We also propose an efficient stochastic auto-encoding variational Bayesian algorithm for the posterior inference and learning for MixSeq. Our experiments on both synthetic and real-world data suggest that MixSeq can capture the characteristics of time series in different groups and improve the forecasting performance of macroscopic time series.

## Acknowledgments and Disclosure of Funding

This work is supported by Ant Group.

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
