# Supplementary materials for Paper "MixSeq: Connecting Macroscopic Time Series Forecasting with Microscopic Time Series Data"

**Zhibo Zhu**[*]
Ant Group
gavin.zzb@antgroup.com

**Ziqi Liu**[*]
Ant Group
ziqiliu@antgroup.com

**Ge Jin**
Ant Group
elvis.jg@antgroup.com

**Zhiqiang Zhang**
Ant Group
lingyao.zzq@antgroup.com

**Lei Chen**
Ant Group
qingli.cl@antgroup.com

**Jun Zhou**[†]
Ant Group
jun.zhoujun@antgroup.com

**Jianyong Zhou**
Ant Group
neil.zjy@antgroup.com

## A Proofs

**Proposition 1.** *Assuming the mixture model with probability density function $f(x)$, and corrresponding components $\{f_i(x)\}_{i=1}^{K}$ with constants $\{p_i\}_{i=1}^{K}$ ($\{p_i\}_{i=1}^{K}$ lie in a simplex), we have $f(x) = \sum_i p_i f_i(x)$. In condition that $f(\cdot)$ and $\{f_i(\cdot)\}_{i=1}^{K}$ have first and second moments, i.e., $\mu^{(1)}$ and $\mu^{(2)}$ for $f(x)$, and $\left\{\mu_i^{(1)}\right\}_{i=1}^{K}$ and $\left\{\mu_i^{(2)}\right\}_{i=1}^{K}$ for components $\{f_i(x)\}_{i=1}^{K}$, we have:*

$$\sum_i p_i \cdot \mathrm{Var}(f_i) \leq \mathrm{Var}(f). \tag{1}$$

*Proof.* We prove the result based on the fact that we have for any moment $k$ that

$$\mu^{(k)} = \mathbb{E}_f\left[x^k\right] = \sum_i p_i \mathbb{E}_{f_i}\left[x^k\right] = \sum_i p_i \mu_i^{(k)}. \tag{2}$$

We then derive the variance of mixture as

$$\mathrm{Var}(f) = \sum_i p_i \mu_i^{(2)} - \left(\sum_i p_i \mu_i^{(1)}\right)^2 = \sum_i p_i \left(\mathrm{Var}(f_i) + \left(\mu_i^{(1)}\right)^2\right) - \left(\sum_i p_i \mu_i^{(1)}\right)^2$$
$$= \sum_i p_i \mathrm{Var}(f_i) + \sum_i p_i \left(\mu_i^{(1)}\right)^2 - \left(\sum_i p_i \mu_i^{(1)}\right)^2. \tag{3}$$

Since the squared function is convex, by Jensen's Inequality we immediately have $\sum_i p_i \left(\mu_i^{(1)}\right)^2 \geq \left(\sum_i p_i \mu_i^{(1)}\right)^2$. □

---

[*]Equal contribution.
[†]Corresponding author.

35th Conference on Neural Information Processing Systems (NeurIPS 2021).

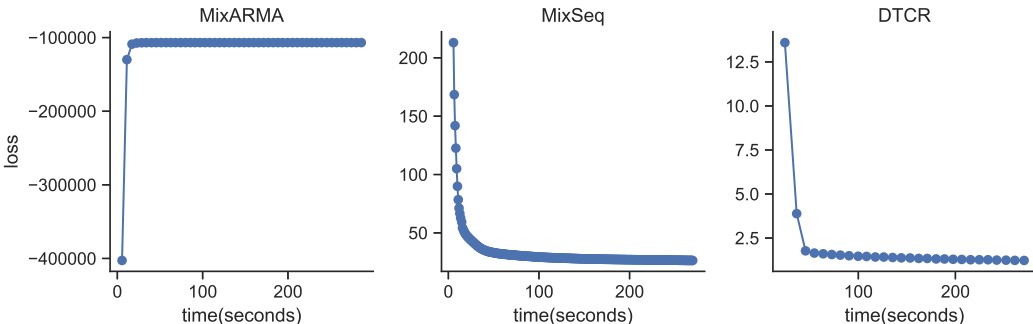

Figure 1: The convergence of different clustering methods over time (seconds) on Rossmann dataset. The optimization objective of MixARMA is maximized by EM algorithm, and the optimization objective of MixSeq and DTCR are minimized by gradient descent with Adam.

## B    Complexity analysis and running time

**Complexity analysis of MixSeq.** The complexity of MixSeq depends on the number of clusters $K$ and three network architectures, including convolution, multi-head self-attention and MLP. For time series $x \in \mathbb{R}^{t \times d}$, where $t$ and $d$ are the length and dimension of time series respectively, the FLOPs (floating point operations) of convolution is $O(w_k \cdot d \cdot t \cdot d_k)$, where $w_k$ and $d_k$ are the size and number of convolution kernel. The FLOPs of multi-head self-attention is $O(h \cdot t^2 \cdot d_k)$, where $h$ is the number of multi-heads. The FLOPs of MLP is $O(h \cdot t \cdot d_k^2)$. Finally, the FLOPs of MixSeq is $O(K(w_k \cdot d \cdot t \cdot d_k + h \cdot t^2 \cdot d_k + h \cdot t \cdot d_k^2))$. Since $w_k$, $d_k$ and $d$ are usually smaller than $t$, so the time complexity can be simplified as $O(K \cdot h \cdot t^2 \cdot d_k)$ which is similar to Transformer. Time series data is always recorded by day or hour. There are only one thousand values even for the data recorded in three years, so our method is capable for dealing with them. Furthermore, some existing methods can also be used in MixSeq to accelerate the computation of self-attention.

**Running time of macroscopic time series forecasting.** The overall running time of forecasting macroscopic data based on MixSeq is comprised of two steps. **(1)** The first step is to do clustering with MixSeq. Figure 1 shows the convergence of MixSeq over time (seconds) compared with comparison approaches to time series clustering. Our approach takes 200 seconds to convergence on the dataset containing $1,115$ microscopic time series, while MixARMA and DTCR take 20 and 200 seconds to convergence respectively. The convergence rate of our method is not worse than existing neural network based approach, i.e., DTCR. **(2)** The second step is to forecast clustered time series with any proper forecasting model. The time complexity is in linear w.r.t the number of clustered time series. We can always accelerate this step by using more workers in parallel.

## C    Evaluation metrics

### C.1    Rand index

Given the labels as a clustering ground truth, Rand index (RI) measures the clustering accuracy between ground truth and predicted clusters, defined as

$$\text{RI} = \frac{a + b}{C_m^2},$$

where $m$ is the total number of samples, $a$ is the number of sample pairs that are in the same cluster with same label, and $b$ is the number of sample pairs that in different clusters with different labels.

## C.2 Symmetric mean absolute percentage error

Symmetric mean absolute percentage error (SMAPE) is an accuracy measure for time series forecasting based on percentage (or relative) errors. $\text{SMAPE} \in [0, 1]$ is defined as

$$\text{SMAPE} = \frac{1}{n} \sum_{t=1}^{n} \frac{|x_t - \hat{x}_t|}{|x_t| + |\hat{x}_t|},$$

where $x_t$ is the actual value and $\hat{x}_t$ is the predicted value for time $1 \leq t \leq n$, and $n$ is the horizon of time series forecasting.

## C.3 $\rho$-quantile loss

In the experiments, we evaluate different methods by the rolling window strategy. The target value of macroscopic time series for each dataset is given as $x_{i,t}$, where $x_i$ is the $i$-th testing sample of macroscopic time series and $t \in [0, 30)$ is the lead time after the forecast start point. For a given quantile $\rho \in (0, 1)$, we denote the predicted $\rho$-quantile for $x_{i,t}$ as $\hat{x}_{i,t}^{\rho}$. To obtain such a quantile prediction from the estimation of clustered time series, a set of predicted samples of each clustered time series is first sampled. Then each realization is summed and the samples of these sums represent the estimated distribution for $x_{i,t}$. Finally, we can take the $\rho$-quantile from the empirical distribution.

The $\rho$-quantile loss is then defined as

$$R_\rho(\mathbf{x}, \hat{\mathbf{x}}^\rho) = \frac{2 \sum_{i,t} D_\rho(x_{i,t}, \hat{x}_{i,t}^\rho)}{\sum_{i,t} |x_{i,t}|}, \qquad D_\rho(x, \hat{x}^\rho) = (\rho - \mathbf{I}_{\{x \leq \hat{x}^\rho\}})(x - \hat{x}^\rho)$$

where $\mathbf{I}_{\{x \leq \hat{x}^\rho\}}$ is an indicator function.

# D  Experiments

## D.1  Enviroment setting

We conduct the experiments on an internal cluster with 8-core CPU, 32G RAM and 1 P100 GPU. Meanwhile, MixSeq, together with the time series forecasting methods based on neural network, are implemented with tensorflow 1.14.0.

## D.2  Simulation parameters of toy examples

We use the TimeSynth[3] python package to generate simulation time series data. For GP time series, we use RBF as the kernel function. The lengthscale and variance are $[1.5, 2]$, $[0.5, 2.5]$ and $[0.5, 1]$ for the mixture model of 3 GPs. Then, we add time series generated by GP with $[0.5, 0.5]$ and $[2, 1]$ for the samples from mixture of 5 GPs. Similarly, we use ARMA(2, 0) to generate ARMA time series. The parameters of the first three components of the mixture of ARMA are $[1.5, -0.75]$, $[1, -0.9]$ and $[-0.25, 0.52]$ respectively. The parameters of another two components are $[0.34, 0.27]$ and $[1, -0.30]$. The initial values of ARMA are sampled from $N(0, 0.25)$.

# E  Societal impacts

We study the problem that whether macroscopic time series forecasting can be improved by leveraging microscopic time series, and finally propose MixSeq to cluster microscopic time series to improve the forecasting of macroscopic time series. This work will be especially useful for financial institutions and e-commercial platforms, e.g., loan forecasting, balance forecasting, and Gross Merchandise Volume (GMV) forecasting. The forecasting can help business decisions like controlling the risk of each financial institution, and help the lending to merchant in an e-commerce platform. Misuse of microscopic data could possibly lead to privacy issues. As such, protecting microscopic data with privacy-preserving techniques should be important.

---

[3]https://github.com/TimeSynth/TimeSynth