# OpenReview forum: "MixSeq: Connecting Macroscopic Time Series Forecasting with Microscopic Time Series Data"
_NeurIPS.cc/2021/Conference — NeurIPS 2021 Poster_

### Official Review · Reviewer_viNa · 2021-07-09

**Rating:** 5
**Confidence:** 3

**Summary:**

This paper tackles the problem of predicting a so called macroscopic time series that is the aggregation of multiple microscopic time series. For example, global sales of a chain can be the sum of sales in individual locations. The authors propose to predict the macroscopic time series by building a mixture model on microscopic time series. They provide a theoretical and empirical justification of this approach. They also compare their approach to baselines on synthetic and 3 real-world data sets. For the latter data sets, performance on microscopic time series is also evaluated. The results suggest the proposed approach is better than the competing baselines.

**Limitations And Societal Impact:**

Yes

**Main Review:**

**Originality**
I think the originality of the paper is good.
- The related work section is informative, well written and broad.
- I am not aware of any work that specifically studies time series forecasting from microscopic time series. I think it is an interesting topic, as it happens frequently.
- While mixture models are not new and the approach proposed here does not bring much novelty, I believe its application to the problem is nevertheless interesting.

**Quality**
I think the quality of the paper is average.
- It is quite possible I am missing something, but I cannot get my head around the theoretical justification for the approach (proposition 1). I would really like the authors to address this and/or correct me in the rebuttal. It states that $\sum_i p_i Var(f_i) \leq Var(f)$, where $f_i$ denotes the distribution over a cluster and the mixture distribution $f(x)=\sum_i p_i f_i(x)$. Based on this, the proposed approach is motivated by the fact that modeling individual clusters could be more accurate than modeling the mixture directly. The demonstration with toy examples is very nice though.
   - My first question is about the demonstration. I agree that the variance of the mixture is higher than the weighted sum of individual variances. However, I am not sure that the moment of any order (k) of the mixture is the weighted sum of the corresponding cluster moments: $\mu^{(k)} = \sum_i p_i \mu_i^{(k)}$   (equation 2 in the appendix).
   - My second issue is that the goal of the paper is not to model the mixture itself, but an aggregation (the macroscopic time series) of multiple realization of the mixture (the microscopic time series). In other words, the variance of interest is not the variance of the mixture, but the variance of a sum of random variables. Hence I do not see the relevance of the theoretical justification.
- The experiments on the other hand are very nice. The baselines seem to make sense, there are three real world and relevant data sets. The proposed approach outperforms the baselines most of the time. They also show the strength of the approach by considering variations such as only modelling the macroscopic time series. I also like that both the macroscopic and the microscopic time series forecasting are considered on the real world data sets as both are likely to be useful in practice. I think it is not necessary for the paper to be good, but it would have been nice to have some exploration of the clusters learned by the model.
- Chosing the number of clusters is not discussed. This is not a big problem as anybody who has worked on mixture model will have faced this problem, but some pointers could be nice for readers not familiar with the topic.

**Clarity**
I think the clarity of the paper is not good.
- The paper is well organized and easy to follow at a high level.
- The experiments are well described. I think they could be reproduced easily.
- However, the paper should be carefully proofread for spelling and grammatical mistakes. There are too many left and this significantly impacts readability. In my opinion in its current state the paper is not ready to be published. This is my main concern with this paper.

**Significance**
I think significance is fine.
- The addressed problem is very specific, but it is encountered in practice.
- The proposed approach beats the baselines very often.

**Time Spent Reviewing:**

3

---

> ### Author Response · Authors · 2021-08-10
> **Responses to reviewer viNa: explanation of proposition; sensitivity analysis, and more experimental results on the analysis on the number of clusters.**
>
> Thanks for your careful and valuable comments. The responses to your concerns are as below.
>
> **Q1: I am not sure that the moment of any order (k) of the mixture is the weighted sum of the corresponding cluster moments: $\mu^{(k)} = \sum_i p_i \mu_i^{(k)}$ (equation 2 in the appendix).**
>
> A1: Given $f(x) = \sum_i p_i f_i(x)$, we have
> $$
> \mu^{(k)} = \mathbb{E} _f \left[ x^k \right] = \int x^k \cdot f(x) dx
> = \int x^k \cdot \sum_i p_i f_i(x) dx = \sum_i p_i \int x^k \cdot f_i(x) dx
> = \sum_i p_i \mathbb{E} _{f_i} \left[ x^k \right] = \sum_i p_i \mu_i^{(k)}
> $$
> Therefore, the raw moment of any order of the mixture is the weighted sum of the corresponding cluster moments.
>
> **Q2: My second issue is that the goal of the paper is not to model the mixture itself, but an aggregation (the macroscopic time series) of multiple realization of the mixture (the microscopic time series). In other words, the variance of interest is not the variance of the mixture, but the variance of a sum of random variables. Hence I do not see the relevance of the theoretical justification.**
>
> A2: We are interested in the variance of the macroscopic time series that is the aggregation of microscopic time series. First, Proposition 1 states that if we knew the ground truth cluster assignment of each microscopic time series, the variance on expectation conditioned on the ground truth cluster assignment should be no larger than the variance of the mixture. Second, based on the assumption that microscopic data are independent, the variance of aggregated microscopic data is just the sum of the variances of microscopic data.
>
> As a result, the variance of the aggregation of clustered time series should be at least no larger than the aggregation of all microscopic time series, i.e., the macroscopic time series. That means first clustering the microscopic data into clustered time series, and followed by standard forecasting for each clustered time series should be better. This motivates us the proposal of MixSeq to clustering microscopic time series, and the reason why macroscopic time series forecasting can be improved.
>
> **Q3: I think it is not necessary for the paper to be good, but it would have been nice to have some exploration of the clusters learned by the model.**
>
> A3: We show in synthetic experiments that given the ground truth mixture components, our approach can well capture and recover the clustered time series. This demonstrates MixSeq’s capability of clustering microscopic time series following various probabilistic mixture distributions. For data without ground truth mixture components known a priori, we would give some explanation based on some measures like the spectrum of frequencies or other quantities in future work. In practice, while forecasting loans for a financial institution, we would analyze the clusters based on users' profiles, the frequency/amount of loans so as to explain the clusters.
>
> **Q4: Choosing the number of clusters is not discussed. This is not a big problem as anybody who has worked on mixture model will have faced this problem, but some pointers could be nice for readers not familiar with the topic.**
>
> A4: We strongly suggest the reviewer could check the sensitivity analysis on the number of clusters in our supplementary file in Figure 2.
>
> We further conduct a new synthetic experiment to show the influence of the number of clusters, i.e., we show if the ground truth number of clusters is 3, the forecasting performance would degenerate in case we do not set this hyperparameter properly. The details are as follows. Following the experimental setting in Section 5.1, we generate 10000 microscopic time series from 3 different parameteried ARMA respectively (30000 in total). The aggregation of all samples is the macroscopic time series which is the forecasting target. Then, we compare the forecasting performance between the method that directly forecasts macroscopic time series (denoted as Macro) and our method with different cluster numbers (including 2, 3, 5,  denoted as MixSeq_2, MixSeq_3, and MixSeq_5 respectively). We apply the rolling window approach with ARMA for T+10 forecasting in the last 40 time steps. The average results (SMAPE, the lower the better) of 5 trials are 0.807, 0.774, 0.731 and 0.756 for Macro, MixSeq_2, MixSeq_3 and MixSeq_5 respectively. Similar to the results in Figure 2 in supplementary, our method is superior to Macro. Meanwhile, MixSeq_3 with the ground truth cluster number has the best performance, which also states the importance of setting the proper number of clusters. We suggest doing binary search on this critical hyperparameter.

---

> ### Comment · Reviewer_viNa · 2021-08-25
> **Thank you for the clarification**
>
> I would like to thank the authors for their clarifications. They have addressed my questions about the theoretical justification of the approach. I think the research described in the paper is good and interesting. However I am still concerned about the amount of language mistake in the original submission. If not for that, I would vote to accept the paper. That being said I am fine with the AC deciding this is no ground for rejection. I wish the authors the best for continuing this line of research.
>
> I will update my score accordingly.

---

> > ### Author Response · Authors · 2021-08-26
> > **Thank you for the response**
> >
> > Thanks for reading our response and giving your feedback. We have proofread the submission and carefully fixed most of the language mistakes. Sincerely, we really hope that this does not affect the final score.

---

> > ### Author Response · Authors · 2021-09-03
> > **About language mistakes**
> >
> > Thanks for the reviewer's feedback again. For the concerns about language mistakes, please see some of our revisions as follows.
> >
> > **L1**. ***Time*** (~~Times~~) series forecasting is widely used in business intelligence...
> >
> > **L27**. ... all of them study the modeling of time series without considering the connections between macroscopic time series of interest and the ***underlying*** (~~corresponding~~) time series in the microscope.
> >
> > **L31**. Basically, though (~~the~~) accurately modeling (~~of~~) each microscopic time series could be challenging due to large variations ***of each*** (~~among large number of~~) microscopic data, we show that by carefully clustering microscopic time series into clusters, i.e., clustered time series, and ***using*** (~~use~~) canonical approaches to model each of clusters, finally we can achieve promising results by simply summing over the forecasting results of each cluster.
> >
> > **L37**. To be more specific, first, we assume that the microscopic time series ***are generated from*** (~~are followed by~~) a probabilistic mixture model where there exist K components.
> >
> > **L44**. Second, inspired by recent ***successes*** (~~success~~) of Seq2seq models based on deep neural networks, e.g., variants of recurrent neural networks (RNNs), convolutional neural networks (CNNs), and Transformers, we propose Mixture of Seq2seq (MixSeq), a mixture model for time series, where the components ***come from*** (~~follow by~~) a family of Seq2seq models parameterized by different parameters.
> >
> > **L66**. We aim to ***cluster*** (~~clustering~~) the m microscopic time series into K clustered time series, ..., given the label assignment of ***the*** $i$-th microscopic time series ...
> >
> > **L68**. This is based on our results in Section 3 that ~~show~~ the macroscopic time series forecasting can be improved with ~~an~~ optimal clustering.
> >
> > **L76**. The encoder feeds $x_{<t}$ into a neural architecture, e.g., RNNs, CNNs ,or self-attentions, to generate ***the*** representation of historical time series, denoted as $h_t$, then ***we use*** (~~uses~~) a decoder to yield the result.
> >
> > **L170**. ... modeling on clustered data aggregated by time series from ***the*** same component would have better results ***compared with*** (~~than~~) modeling the macroscopic time series.
> >
> > **L218**. ... the KL divergence has few ***effects*** (~~effect~~) on the optimization objective...
> >
> > **L242**. ... to the parameters of ***the*** base model.
> >
> > **L268**. Then, we train the forecasting ***models*** (~~model~~) on the clustered time series separately...
> >
> > **L283**. We also report the ***results (named as Macro)*** (~~performance~~) that ***we*** directly ***train*** (~~building~~) forecasting ***models*** (~~model~~) on the macroscopic time series without (~~performing~~) leveraging the microscopic data (~~, named as “Macro”~~).
> >
> > **L299**. Regarding time series forecasting ***models*** (~~model~~), we apply the default setting to ARMA and Prophet provided by ***the*** Python ***packages*** (~~package~~).
> >
> > We **promise** to carefully address the typos and language mistakes, and we will carefully revise this submission according to reviewers' comments. If we have addressed the reviewer's concerns, we really appreciate it if the reviewer could vote to accept.

---

### Official Review · Reviewer_pibM · 2021-07-10

**Rating:** 6
**Confidence:** 3

**Summary:**

Authors model time series as a weighted mixture of a set of time series for forecasting. Proposed a model for Mixture of seq2seq model. Number of components depends on correct clustering of micro time series. The paper also  propose an efficient stochastic auto-encoding variational Bayesian algorithm for the posterior inference and learning for MixSeq


**Ethics Review Area:**

["Inadequate Data and Algorithm Evaluation"]

**Limitations And Societal Impact:**

Many real world data need to forecast which are important for damage management. Author can discuss if the proposed forecasting method can handle forecasting such events like extreme rain fall. Wrong forecasting may lead to lots of damage of human life and properties.

**Main Review:**

The idea of using a mixture of models in time series prediction is already there [30, 39], and Proposition-1 is about any general data point and function which is known. The first contribution seems incremental. The result shows accuracy increases monotonically with the splitting of the time-series into components. There is no result or discussion then how many components are sufficient or it is really monotonic. Then how do we choose the number of components?
 The second contribution of building up the mixture model of seq2seq is new and the learning process is original. However, the use of seq2seq over RNN is not very clear.

The Significance  of the task of forecasting time series data is well known. The paper could not demonstrate the advantage of the proposed method. Theoretical result is not presented regarding this and more over Empirical evaluation is not significant.

I have following question regarding Empirical set up  and claims
Various tables report various metrics without any proper discussion. It will be better to have consistency over comparing method and performance metric.
Sensitivity analysis on hyperparameters is not presented. It seems the author has used validation to fix hyperparameters.  But how sensitive is model performance to the number of cluster assumptions.
Numbers reported in table 5 are very much comparable [{arma: 0.0269(0.0002) seq:0.0267(0.0002)};{0.2736(0.0013) 0.2733(0.0012)}]. Is there any statistical significance?

Paper is not easy to read. Did not have discussion of state of art mixture models for time series forecasting.


**Time Spent Reviewing:**

8

---

> ### Author Response · Authors · 2021-08-10
> **Responses to reviewer pibM: misunderstood of our contribution; sensitivity analysis, and metrics; more experiments on the analysis of the number of clusters.**
>
> Thanks for your careful and valuable comments. The responses to your concerns are as below.
>
> **Q1: The idea of using a mixture of models in time series prediction is already there [30, 39], and Proposition-1 is about any general data point and function which is known. The first contribution seems incremental. The result shows accuracy increases monotonically with the splitting of the time-series into components. There is no result or discussion then how many components are sufficient or it is really monotonic. Then how do we choose the number of components?**
>
> A1: We did a thorough review of existing time series clustering approaches, and show that none of them aim at forecasting macroscopic time series, and how to leverage the results of time series clustering for improving forecasting remains an open question. Please check the related works and their experiments (no results on forecasting)! Hence we propose MixSeq for the purpose of forecasting. This is fundamentally different from existing time series clustering approaches with the purpose of clustering and analysis. We still believe that this is the first work in the literature that studies why the clustering of microscopic time series can help forecast macroscopic time series and how we can do it. We also do not agree that the state space model in [30] can be viewed as a mixture time series model.
>
> We have to point out the factual error that the reviewer misunderstood: "the accuracy increases monotonically with the splitting of the time series into components". Our result in Proposition 1 states that if we knew the ground truth cluster assignment of each microscopic time series, the variance on expectation conditioned on the ground truth cluster assignment should be no larger than the variance of the mixture. Based on the assumption that microscopic data are independent, the variance of the aggregation of clustered time series should be at least no larger than the aggregation of all microscopic time series, i.e., the macroscopic time series. In other words, the forecasting of clustered time series could be no worse than forecasting the macroscopic time series directly. However, when we cannot appropriately clustering the microscopic time series, our results would degenerate.
>
> We strongly ask the reviewer could have a look at our supplementary file in D.3 where Figure 2 shows the forecasting performance with various numbers of clusters, i.e., sensitivity analysis, given that the reviewer has missed. We suggest doing binary search on this critical hyperparameter.
>
> We further setup a new synthetic experiment to show the degeneration of forecasting performance when we set the number of clusters inappropriately. The details are as follows. We set the ground truth number of clusters as 3. Following the experimental setting in Section 5.1, we generate 10000 microscopic time series from 3 different parameteried ARMA respectively (30000 in total). The aggregation of all samples is the macroscopic time series which is the forecasting target of interest. Then, we compare the forecasting performance between the method that directly forecasts macroscopic time series (denoted as Macro) and our method with different cluster numbers (including 2, 3, 5,  denoted as MixSeq_2, MixSeq_3, and MixSeq_5 respectively). We apply the rolling window approach for T+10 forecasting in the last 40 time steps. The average results (SMAPE, the lower the better) of 5 trials are 0.807, 0.774, 0.731 and 0.756 for Macro, MixSeq_2, MixSeq_3 and MixSeq_5 respectively. MixSeq_3 being set with ground truth cluster number shows the best performance, while MixSeq_2 and MixSeq_5 would degenerate even though they still do better than Macro. This result shows the importance of setting a proper number of clusters.
>
> **Q2: The second contribution of building up the mixture model of seq2seq is new and the learning process is original. However, the use of seq2seq over RNN is not very clear.**
>
> A2: We formulate the encoder and decoder of MixSeq using a Seq2seq architecture, especially instantiating it with a Transformer (Vaswani, Ashish, et al. "Attention is all you need." NeurIPS. 2017) instead of an RNN architecture. We understand Seq2seq as a general arch but not limited to RNN. In the meanwhile, we would like to test RNN in MixSeq as an alternative in future works, even though it is well known that RNN is difficult to handle long time series. However, this should not be the major contribution of this work.
>
> **Q3: Various metrics without any proper discussion. It will be better to have consistency over comparing method and performance metric. Sensitivity analysis on hyperparameters is not presented. It seems the author has used validation to fix hyperparameters. But how sensitive is model performance to the number of cluster assumptions.**
>
> A3: We strongly suggest the reviewer could have a look at our supplementary file and check those metrics. Basically, SMAPE and ρ-quantile loss are used to evaluate the accuracy of time series forecasting. Rand Index is used to evaluate the accuracy of recovering ground truth cluster assignment in case we know the ground truth in synthetic data.
>
> We also strongly suggest the reviewer could have a look at our supplementary file and check the sensitivity analysis of hyperparameters, i.e., number of clusters. Doing binary search over this hyperparameter and setting a proper number is critical. We also add synthetic experiments as discussed above to show that setting an improper number could degenerate the forecasting performance. Please check!
>
> **Q4: Many real world data need to forecast which are important for damage management. Author can discuss if the proposed forecasting method can handle forecasting such events like extreme rain fall. Wrong forecasting may lead to lots of damage of human life and properties.**
>
> A4: We do not ever claim that our approach can handle the forecasting of extreme events where there exists no connection between macroscopic time series and microscopic time series. Instead, we study the forecasting of macroscopic time series by leveraging microscopic data. Such forecasting problems are important, e.g., the loans of a financial institution that is aggregated from the loans of each customer, the sales from an online retail platform which is comprised of the sales of each merchant. The forecastings can help the business decision of financial institutions etc.

---

### Official Review · Reviewer_af7g · 2021-07-15

**Rating:** 7
**Confidence:** 3

**Summary:**

This paper develops methodology for improving time series forecasting by modeling finer-scale structure in the time series; namely, the underlying "microscopic" time series of which the predicted "macroscopic" time series is aggregated. The approach takes advantage of the concept that in many cases the time series of interest are in fact macroscopic aggregates of smaller-scale structure and time series. The work develops a method that models the micro time series as a mixture of latent components and then uses a neutral network based approach to model and parameterize the mixture. Benchmarkins are done with simulated and real data, showing promising performance.

**Limitations And Societal Impact:**

According to the checklist, the authors have not addressed the limitations and societal impacts.

Suggestion for improvement would be to go through the checklist and address the required points in the discussion. Further evaluation is possible after this.

**Main Review:**

The originality relates to bringing together the underexplored idea of aggregating information across microscopic time series, and a well-motivated modeling approach. The methodology appears justified and sound and the reporting is clear. The main significance of this work is in opening up a new line of analysis for micro time series aggregation based on neural & autoencoder models, for improving time series forecasts.

Strengths:
+ Thorough analysis
+ Pragmatic idea
+ Clear performance improvements

Weaknesses:
- Not clear if all topics in the checklist have been properly addressed (several "No" questions in the form)
- More information / examples on common settings where the assumptions break and alternative models are better would be useful
- More information on running times would be useful



**Time Spent Reviewing:**

3

---

> ### Author Response · Authors · 2021-08-10
> **Responses to Reviewer af7g: the assumption break, and running time.**
>
> Thanks for your careful and valuable comments. The responses to your concerns are as below.
>
> **Q1: More information / examples on common settings where the assumptions break and alternative models are better would be useful.**
>
> A1: Our Proposition 1 states that if we knew the ground truth cluster assignment of each microscopic time series, the variance on expectation conditioned on the ground truth cluster assignment should be no larger than the variance of the mixture. Based on the assumption that microscopic data are independent, the variance of the aggregation of clustered time series should be at least no larger than the aggregation of all microscopic time series, i.e., the macroscopic time series. So estimating ground truth clustered time series should be at least no worse than forecasting macroscopic time series directly. In the case that the data have only one mixture component, our result reduces to estimate macroscopic time series, i.e., a tie in this case.
>
> Empirically, when we cannot recover appropriate clusters over microscopic time series, we will lose above guarantees. Our method would be useful by calibrating a proper number of components, i.e., the number of clusters. Figure 2 in supplementary shows the forecasting performance with various numbers of clusters. We suggest doing binary search on this critical hyperparameter.
>
> We further setup a new synthetic experiment to show the degeneration of forecasting performance when we set the number of clusters inappropriately. The details are as follows. We set the ground truth number of clusters as 3. Following the experimental setting in Section 5.1, we generate 10000 microscopic time series from 3 different parameteried ARMA respectively (30000 in total). The aggregation of all samples is the macroscopic time series which is the forecasting target of interest. Then, we compare the forecasting performance between the method that directly forecasts macroscopic time series (denoted as Macro) and our method with different cluster numbers (including 2, 3, 5,  denoted as MixSeq_2, MixSeq_3, and MixSeq_5 respectively). We apply the rolling window approach for T+10 forecasting in the last 40 time steps. The average results (SMAPE, the lower the better) of 5 trials are 0.807, 0.774, 0.731 and 0.756 for Macro, MixSeq_2, MixSeq_3 and MixSeq_5 respectively. MixSeq_3 being set with ground truth cluster number shows the best performance, while MixSeq_2 and MixSeq_5 would degenerate though still better than Macro. This result shows the importance of setting a proper number of clusters.
>
> **Q2: More information on running times would be useful.**
>
> A2: The overall running time of forecasting macroscopic data is comprised of two steps.
>
> The first step is to do clustering with MixSeq. Figure 1 in supplementary material shows the convergence of MixSeq over time (seconds) compared with comparison approaches to time series clustering. Our approach takes 200 seconds to convergence on the dataset containing 1115 microscopic time series, while MixARMA [39] and DTCR [21] take 20 and 200 seconds to convergence respectively. The convergence rate of our method is not worse than existing neural network based approach, i.e., DTCR.
>
> The second step is to forecast clustered time series with any proper forecasting model. The time complexity is in linear w.r.t the number of clustered time series. We can always accelerate this step by using more workers in parallel.
>
> **Q3: Not clear if all topics in the checklist have been properly addressed (several "No" questions in the form).**
>
> A3: We will carefully go through the checklist and update them in the subsequent material.

---

> ### Author Response · Authors · 2021-08-26
> **More clarifications about checklist.**
>
> Thanks for the reviewer's valuable comments again.
>
> We have some further clarifications about the checklist.
> 1. "license and new assets". All the real-world data are public. We will release the source code.
> 2. "limitations". We hope our answer to "the assumptions break" can appropriately address this issue.
> 3. "societal impact". This work will be especially useful for financial institutions and e-commercial platforms, e.g., loan forecasting, balance forecasting, and Gross Merchandise Volume (GMV) forecasting. The forecasting can help business decisions like controlling the risk of each financial institution, and help the lending to merchant in an e-commerce platform. Misuse of microscopic data could possibly lead to privacy issues. As such, protecting microscopic data with privacy-preserving techniques should be important.
>
> If we have addressed your concerns, we really appreciate that you could remain the original score unchanged.

---

### Decision · Program_Chairs · 2021-09-27

**Decision:**

Accept (Poster)

**Comment:**

The paper proposes an approach for forecasting a single "macroscopic" time series which is the sum of several "microscopic" time series by forecasting the microscopic time series using a mixture of transformer-based seq2seq models and finally combining the forecasts.

The reviewers agree that the paper proposes a pragmatic solution to a relevant problem, and demonstrates clear performance improvements in doing so. While the building blocks (mixture models trained with variational inference, ConvTrans time series model) are well know, the combination is novel and well-chosen to address the task at hand. The reviewers highlight the convincing empirical evaluation against sensible baselines, where significant performance gains were demonstrated. Initial reviewer concerns around novelty and clarity were alleviated during the discussion period, leading me to recommend acceptance.